# Exosomes in Systemic Sclerosis: Messengers Between Immune, Vascular and Fibrotic Components?

**DOI:** 10.3390/ijms20184337

**Published:** 2019-09-04

**Authors:** Marta Colletti, Angela Galardi, Maria De Santis, Giacomo Maria Guidelli, Angela Di Giannatale, Luigi Di Luigi, Cristina Antinozzi

**Affiliations:** 1Department of Pediatric Hematology/Oncology, IRCCS, Ospedale Pediatrico Bambino Gesù, Viale San Paolo 15, 00146 Rome, Italy; 2Rheumatology and Clinical Immunology, Humanitas Clinical and Research Center-IRCCS—Via Alessandro Manzoni, 56, 20089 Rozzano, Milan, Italy; 3Unit of Endocrinology, Department of Movement, Human and Health Sciences, Università degli Studi di Roma “Foro Italico”, Piazza Lauro de Bosis, 600135 Rome, Italy

**Keywords:** systemic sclerosis therapies, rheumatic disease, exosomes, miRNA

## Abstract

Systemic sclerosis (SSc) is a rare autoimmune disease, characterized by vasculopathy and fibrosis of the skin and internal organs. This disease is still considered incurable and is associated with a high risk of mortality, which is related to fibrotic events. An early diagnosis is useful for preventing complications, and targeted therapies reduce disease progression and ameliorate patients’ quality of life. Nevertheless, there are no validated biomarkers for early diagnosis with predictive prognostic value. Exosomes are membrane vesicles, transporting proteins and nucleic acids that may be delivered to target cells, which influences cellular behavior. They play important roles in cell–cell communication, both in physiological and pathological conditions, and may be useful as circulating biomarkers. Recent evidences suggest a role for these microvesicles in the three main aspects related to the pathogenesis of SSc (immunity, vascular damage, and fibrosis). Moreover, exosomes are of particular interest in the field of nano-delivery and are used as biological carriers. In this review, we report the latest information concerning SSc pathogenesis, clinical aspects of SSc, and current approaches to the treatment of SSc. Furthermore, we indicate a possible role of exosomes in SSc pathogenesis and suggest their potential use as diagnostic and prognostic biomarkers, as well as therapeutic tools.

## 1. Introduction

Systemic sclerosis (SSc) is a rare systemic autoimmune disease with an unknown etiology, and it is characterized by a progressive fibrotic process that affects the skin, microvasculature, and numerous internal organs. The pathogenesis of SSc is complex and not fully understood, and it may manifest with multiple different phenotypes, which make it difficult to classify patients and choose the most appropriate treatment. The early and accurate diagnosis and classification of SSc patients may facilitate the timely recognition of life-threatening complications and initiation of targeted therapies to halt their progression. Despite intensive investigations aimed at the identification of outcome measures for SSc, no validated biomarkers exist for the early diagnosis and assessment of SSc disease activity with a predictive prognostic value. In this context, the discovery of biomarkers that identify patient subgroups is of great scientific interest. Exosomes, nanometer-sized membrane vesicles, are natural nanocarriers that play a central role in cell–cell communication, regulating a broad range of physiological and pathological cellular processes. A large body of evidence suggests that miRNAs and proteins are selectively packaged in exosomes that differ in terms of their physiological and pathological conditions. Therefore, these vesicles are studied as potential biomarkers for diagnostic purposes, monitoring the disease evolution and treatment responses. In recent years, these fascinating vesicles have also emerged as natural vehicles for the delivery of drugs and small molecules, such as proteins and nucleic acids, both for increasing bioavailability and for precision medicine. In this review, we describe the latest information on the main aspects of SSc and recent treatment approaches. We also highlight a potential pathogenic role of exosomes in SSc and their possible use for specific drug delivery.

## 2. Systemic Sclerosis

SSc affects about 20 in one million subjects per year [1,2] and patients are predominantly females, with an overall female to male ratio of over 3:1 [3]. In 1980, the American College of Rheumatology (ACR) divided SSc into two classes, based on the extent of skin involvement, as follows: Diffuse systemic sclerosis (dSSc) and limited systemic sclerosis (lSSc) [4]. Environmental and genetic factors can occur in disease pathogenesis, although its exact origin, mechanisms, and etiology are largely unknown. Clinical symptoms include the following three events: A dysregulated immune response to unknown triggers in genetically predisposed subjects, endothelial damage and fibro-proliferative vasculopathy, and fibroblast dysfunction, which can cause an excessive collagen accumulation in the skin and internal organs. These processes interact and affect each other, contribute to disease progression, and lead to an end fibrotic stage and atrophy of the affected organs.

### 2.1. Immune System Abnormalities 

The role of both innate and adaptive immune systems is widely accepted in disease pathogenesis [5], and it is also indicated by the presence of inflammatory processes at the disease onset and the presence of a T-cell, macrophage (M), mast cell, and B-lymphocyte infiltrates in SSc skin patients [6,7]. However, the humoral response against nuclear enzymes, such as topoisomerase or RNA polymerase, together with the endothelial damage related to the Raynaud’s phenomenon (RP), characterizing 99% of SSc patients before the disease onset, seem to be the earliest events [8]. 

B cells are considered to be involved mainly in autoantibody production, even if there is no evidence proving the direct causal significance of SSc autoantibodies in the disease. The presence of autoantibodies is helpful in establishing the diagnosis and predicting the severity and disease progression of SSc [9], which are different in patients with a diffuse or limited disease. For example, patients with a diffuse form of SSc generally have the anti-Scl-70 or anti-topoisomerase antibodies, which are also associated with pulmonary fibrosis, while anti-centromere antibodies are present in 80–90% of patients with a limited form of SSc and are associated with pulmonary arterial hypertension [10]. Other autoantibodies, such as anti-endothelial cell autoantibodies and anti-fibroblast antibodies, have also been detected in patients with a limited or diffuse SSc and are implicated in the apoptosis of endothelial cells and fibroblast activation [11,12]. While several studies demonstrate a significant correlation between the presence of specific antibodies and clinical symptoms, it is not clear that these antibodies have a direct pathogenic role or are only markers of specific disease manifestations [11]. Moreover, B cells seem to be also involved in the production of interleukin (IL)-6 and profibrotic mediators, and importantly, they act as antigen-presenting cells for T cells. Thus, since the immune system is considered a key stimulus for SSc onset, vascular injury and fibrosis, current and novel therapies are targeting immunological activation. This approach seems to be particularly effective in patients with a short disease duration, when inflammation is more intense [13,14,15], given the results of a B-cell depleting therapy [16] and the use of anti-IL-6 antibodies. B cells are, in fact, largely involved in fibrogenesis [6] and vascular damage, and they also produce profibrotic mediators, such as IL-6 and IL-10, and thus promote a T helper (Th) 2 and M2 response [17]. 

Th cells are largely involved in SSc development, with a Th1 and Th17 predominance in tissues affected early [7,8,18]. During fibrosis progression, inflammation decreases, and a Th2-predominant profile occurs [19,20]. Increased levels of the Th2 cell-derived cytokines, IL-4, IL-10, and IL-13, were observed in SSc tissue undergoing fibrosis [21,22]. IL-4 increases collagen production in fibroblasts and transforms growth factor beta (TGF-β) production. TGF-β stimulates the synthesis of proteoglycans and fibronectin, inhibits extra-cellular matrix (ECM) degradation by decreasing the synthesis of matrix metalloproteinases (MMPs), and increases the synthesis of the tissue inhibitor of MMPs [6]. Type 2 immune responses involve other innate players, such as innate lymphoid type 2 cells, M2c CD163^+^ macrophages, basophils, mast cells, eosinophils, and epithelial cells producing IL-33 [23].

Based on an interesting hypothesis, the clonal expansion of sel-reactive CD4^+^ cytotoxic T cells could be responsible for “cytolic fibrosis,” characterized by an induced apoptosis of normal tissue, production of profibrotic cytokines, in particular TGF-β and IL-1β, finally leading to abnormal tissue repair and fibrosis [23].

### 2.2. Vasculopathy

Vascular damage occurs early in SSc and precedes fibrosis. At the microcirculation level, RP and nailfold capillary enlargement are the first alterations, and a low capillary density, avascular areas, and skin ulcers (SUs) occur subsequently [24,25]. RP is the clinical manifestation of the cold-induced vasospasm of the fingers, nose, nipples, and toes and generally represents the first sign of SSc [7,26,27,28]. Different factors released by endothelial and immune cells [22], or neuronal and hormonal mediators [29], have been implicated in RP establishment. Endothelial cells release endothelin-1 (ET-1), an important vasoconstrictor, which is most often found in SSc patients [30,31,32]. Pericytes express platelet-derived growth factor (PDGF)-α and -β receptors, antinuclear antibodies, and the regulator of the mediating vessel maturation of G-protein signaling (RGS-5) [33,34,35]. Vascular smooth muscle cells overexpress adrenoreceptors that intensify the vasoconstrictive response induced by cold stimuli [28]. The deregulation of these factors provokes an overstated vessel constriction in response to neurotransmitters and their receptors, as well as a reduced response to vasodilatory signals [36,37]. Furthermore, in SSc lesional tissues there is a lower level of nitric oxide synthase (eNOS) gene expression and nitric oxide (NO) release [38,39]. An impairment of NO release also contributes to the pathogenesis of SSc, increasing platelet aggregation, vascular smooth muscular cell proliferation, and TGF-β and PDGF release. 

SUs can develop in up to 50% of SSc patients and appear as skin lesions, usually localized in acral areas, and digital ulcers (DUs) occurring on the hands, toes, fingertips, and nails [40]. SSc- structural nailfold capillary alterations consist of a progressive decrease in the density of the capillaries, microhemorrhages, and giant capillaries [28,41]. In patients with RP, the presence of capillary alterations, together with the presence of SSc-related autoantibodies (anti-centromere and anti–topoisomerase I autoantibodies), portend the evolution of RP and help to define scleroderma [42]. Three different patterns can be identified in SSc patients, according to microvascular changes [43], as follows: An “early pattern” (relatively well-preserved capillary distribution, few giant capillaries, few capillary microhemorrhages, and no evident loss of capillaries), an “active pattern” (moderate loss of capillaries, numerous giant capillaries, numerous capillary microhemorrhages, mild disorganization of the capillary architecture, and absent or mild ramified capillaries), and a “late pattern” (severe loss of capillaries, with wide avascular areas, small number of giant capillaries, microhemorrhages, and an intense disorganization of the normal capillary array) [41,43]. Nailfold capillary changes could be a useful biomarker of disease activity and severity [43,44] and correlate with cardiac disease and pulmonary arterial hypertension (PAH) onset [45,46]. 

Vascular alterations also occur in other organs in SSc patients and result in heart and pulmonary dysfunctions, renal crisis (SRC), and gastrointestinal dysfunctions [28,45,46]. PAH is the major cause of death in SSc patients, together with pulmonary fibrosis [47]. It can be an isolated tardive complication of lSSc, with anti-centromere antibodies, or a secondary manifestation-related pulmonary fibrosis [48].

Hearth disease can occur as a primary consequence of SSc, or it can be a secondary manifestation of pulmonary and kidney diseases. It manifests as myocardial damage, vessel fibrosis, and pericardial and valvular damage [49]. Cardiac dysfunctions could be clinically silent until late complications occur; thus, together with pulmonary diseases, they may represent one of the major causes of morbidity and mortality in SSc patients [28]. SRC occurred in 5% of SSc patients in previous cohorts, particularly in the first year of the disease, in the diffuse form, and in the case of steroid therapy [50]. SRC is characterized by severe hypertension, followed by glomerulosclerosis, which ends with acute or chronic renal dysfunction [28], but it no longer represents one of the main complications associated with SSc, thanks to angiotensin converting enzyme (ACE)-inhibitor treatment [28].

Vasculopathy is mainly caused by an abnormal balance between vasculogenesis and angiogenesis, which are both defective in SSc patients. However, the exact mechanisms underlying the vascular changes remain largely unknown. At the histological level, SSc is characterized by a gradual loss of small vessels, dilation of capillaries, and stenosis of arterioles and small arteries. In SSc patients, vascular damage is progressive, and it is followed by defective vascular remodeling and angiogenesis. SSc patients show an overexpression of pro-angiogenic factors, like the vascular endothelial growth factor (VEGF), ET-1, and adhesion molecules, together with elevated levels of anti-angiogenic factors, like angiostatin, C-X-C motif ligand (CXCL)-4 chemokine, and IL-4 cytokine [51,52,53]. These markers are found to be altered in SSc patients from the earliest stages of the disease, and they increase when fibrosis occurs and contribute to vascular injury and disease progression [28,53]. At the same time, a lower expression of the main molecules involved in vasculogenesis is reported (i.e., endothelial progenitor cells and the cysteine-rich heparin binding protein). On the one hand, abnormal neovascularization induces vascular destruction, followed by tissue hypoxia and fibroblast activation (destructive vasculopathy); on the other hand, it induces the occlusion of arterioles and small arterioles, mainly caused by the significant proliferation of endothelial cells and smooth muscle cells (proliferative obliterans vasculopathy) [10]. Animal models only partially represent the vascular defects of SSc. Heterozygous mice with the transcription factor friend leukemia integration (Fli) 1 show vascular fragility, an increase of vessel lesions, a lower platelet/endothelial cell adhesion molecule (PECAM-1) expression, a higher MMP-9 expression, capillary dilatation, and a stenosis of arterioles [54]. Knock-out mice with the scaffold protein Caveolin-1 develop PAH and dermal fibrosis [55]. Heterozygous mice with the zinc finger protein kruppel like factor 5 (Kfl5) show lower blood and pulmonary blood vessels and tissue hypoxia [56]. Recently, Kfl5^+/−^ Fli^+/−^ has been developed in double-heterozygous mice. In these mice, immune alterations, vascular damage, and fibrosis occur spontaneously, mirroring the normal course of the SSc disease [56].

### 2.3. Fibroblast Dysfunction 

Fibrosis is a late event and clinical characteristic of SSc, and it is the major cause of mortality in the disease. It occurs early in the skin but can involve all the organs [57]. Fibrosis is due to an excessive fibroblast activation that induces a significant ECM deposition and leads to organ dysfunction. The main actors are the myofibroblasts, specialized fibroblasts, synthetizing collagen, and other ECM components, and they are a major source of TGF-beta during the fibrotic response [8]. Under normal conditions, myofibroblasts can be transiently detected in injured tissues being rapidly removed by apoptosis [58]. Conversely, in pathological fibrogenesis, myofibroblasts persist, becoming excessive and inducing collagen synthesis and ECM accumulation, ending with the development of chronic scarring [58,59]. In SSc, myofibroblast trans-differentiation occurs in response to TGF-β over-secretion. In these cells, there is a high expression of alpha-smooth muscle actin (SMA) fibers, an extra-domain variant form of fibronectin (Fn^EDA^), which elicits potent toll-like receptor (TLR)-dependent fibrotic responses and maintains fibroblast activation [60], a lower CD-34 antigen expression, and a high expression of the antigen Human Thy-1 (CD90) [61,62]. CD-90 is a glycosylphosphatidylinositol-anchored adhesion molecule [63,64], which is involved in cell–cell adhesion and the recruitment of inflammatory cells to activate microvascular endothelial cells [65]. The serum levels of CD90 are significantly increased in SSc patients, particularly in patients with pulmonary fibrosis and PAH [66].

TGF-β is an important regulator of physiological fibrogenesis for lesion healing and tissue repair, and its deregulated function and the deregulated expression of TGF-β-linked genes have been documented in fibrotic tissues of SSc patients and correlate with the disease activity [8,67]. TGF-β is secreted by different immune cells and induces the profibrotic response, activating different pathways, which involve a small mother, against decapentaplegic (SMAD) proteins, mitogen-activated protein kinases (MAPKs), phosphoinositide 3-kinases (PI3Ks), the calcium-dependent phosphatase calcineurin, and the tyrosine kinase c-ABL [8]. The SMAD pathway is considered the main signal that is deregulated in SSc fibroblasts and contributes to the initiation and development of the pathological fibrotic response [8]. Aside from TGF-β, other studies consider other soluble elements. The connective tissue growth factor is a cysteine-rich protein that is up-regulated by apoptotic endothelial cells (ECs) and regulates cell differentiation, proliferation, apoptosis, and ECM synthesis [28,68]. The PDGF, which is high in SSc patients and is secreted by activated platelets, macrophages, and injured ECs [8,28], works as a mitogenic factor, recruiting fibroblasts to lesional tissues and inducing fibroblast activation, collagen synthesis, and TGF-beta1 synthesis [28]. Monocyte chemoattractant protein-1 and -3 are elevated in the lesional skin and cultured fibroblasts of SSc patients, particularly in the early stage of the disease [8], and they contribute to the attraction of inflammatory cells to lesional skin and the promotion of extracellular matrix protein synthesis in SSc fibroblasts [69]. All of these factors have an important role in fibrotic progression and represent further potential specific candidates for therapeutic approaches.

The skin and lungs are the main organs affected by SSc and lung disease is the most severe complication associated with SSc, leading to death [1,47]. In SSc patients, lung disease is significantly heterogeneous. However, it is possible identify two major pathological processes, as follows: The vasculopathy of medium and small pulmonary vessels, which cause PAH and interstitial fibrosis [2]. 

It seems that an inflammatory phase, named alveolitis, precedes lung fibrosis, which can start as unusual interstitial pneumoniae (NSIP), considered a potential reversible or stoppable phase, and then evolve into usual interstitial pneumoniae (UIP), with an irreversible honeycombing-like appearance, leading to restrictive lung syndrome and end-stage lung disease [70].

## 3. Treatments of Systemic Sclerosis 

Different treatment approaches are offered to ameliorate patients’ quality of life and survival. However, there is no treatment that has been demonstrated to modify the overall disease course [2]. The European League Against Rheumatism (EULAR) and the EULAR Scleroderma Trials and Research (EUSTAR) group have published 16 updated recommendations regarding the pharmacological treatment of SSc. These guidelines consist of 16 articles that address the treatment of several SSc-related organ complications, as follows: RP, DUs, PAH, skin and lung disease, scleroderma renal crisis, and gastrointestinal disease [71].

RP is generally the first symptom of systemic sclerosis and precedes skin fibrosis. Moreover, about 50% of patients with RP also manifest DUs and critical digital ischemia [30]. Some of the therapeutic approaches widely used for RP, with or without DUs, provide vasodilators as calcium channel blockers, NO, prostacyclin analogues, phosphodiesterase inhibitors, and endothelin antagonists [28,72]. These drugs act on blood vessels, modulating vascular smooth muscle relaxation, stabilizing the second messengers of cyclic guanosine monophosphate (cGMP) and cyclic adenosine monophosphate, or inhibiting calcium release. As a result, these drugs significantly reduce the severity of vascular damage in patients with SSc [73], reduce the frequency and intensity of ischemic attacks, improve capillary blood flow, and decrease digital ulcers and pain [74,75,76]. Other therapies that are largely used for the treatment of SSc vasculopathy include bosentan, an endothelin inhibitor, previously approved for PAH treatment [77,78]. 

Treatments for SSc-related interstitial lung diseases (ILDs) mainly involve the use of cyclophosphamide, since the positive results from two high-quality randomized control trials and studies on its tolerable toxicity profile were published [71,79]. Other immunosuppressive treatments used for ILDs include the use of mycophenolate or azathioprine [2], and the latter is particularly used as a maintenance therapy. Mycophenolate mofetil (MMF), an inactive prodrug of mycophenolic acid, which inhibits inosine monophosphate dehydrogenase and the proliferation of T and B cells, has been tested for its capacity to treat SSc-associated ILDs. Several studies have shown that MMF improved the skin score and stabilized the pulmonary function of patients [80,81,82]. Tocilizumab (TCZ), a humanized monoclonal IgG1 antibody that acts by blocking the signals mediated by IL-6, has been proposed as a possible treatment for some manifestations of SSc, including ILD. Interestingly, a recent open-label pilot study showed a skin score improvement and lung function stabilization after TCZ treatment [83]. EULAR guidelines suggest the use of immunosuppressive drugs to improve skin damage in SSc patients. Different reports, with poor randomized control trials, demonstrate the beneficial effects of methotrexate, cyclophosphamide, mycophenolate mofetil, and azathioprine for the treatment of skin manifestations due to early diffuse SSc. However, if immunosuppressive therapy seems positive in the early stage of the disease, when active inflammation occurs, the later fibrotic disease might not respond to immunosuppressants alone. Other treatment approaches address the use of corticosteroids, frequently prescribed for patients with diffuse skin involvement or with ‘overlap’ clinical features. However, some studies prove the association between the use of corticosteroids and the serious complication of SRC [84], suggesting their potentially dangerous effects as active scleroderma skin disease treatments. In a recent trial, nintedanib was proven to significantly lower lung volume decline in SSC-related lung fibrosis [85].

Current therapies for PAH include the use of oral agents, such as endothelin receptor antagonists (bosentan and macitentan), phosphodiesterase type 5 inhibitors (sildenafil and taldalafil), and a cGMP inhibitor (riociguat), in particular for moderate and severe PAH, while a continuous intravenous infusion of a prostacyclin analogue (epoprostenol and treprostinil) is used for worse cases [2]. Different studies and clinical trials report that the exercise capacity and some hemodynamic measures are improved in SSc–PAH patients treated with these drugs [86,87,88,89,90,91]. While current therapies may ameliorate survival, there have been no positive results of long-term treatments, especially for worse lung disease conditions [92,93,94]. To prevent disease progression, combination therapies of vasodilator drugs with immunosuppressants and antifibrotic agents have emerged as a new treatment strategy [2]. 

No clinical trials are available for the treatment of renal crisis and SSc-related gastrointestinal disease. However, clinical experts suggest the use of ACE inhibitors for renal crisis, proton pump inhibitors to prevent gastro-esophageal reflux disease (GORD), esophageal ulcers and strictures, and prokinetic drugs for motility disturbances (i.e., dysphagia, GORD, early satiety, bloating, and pseudo-obstruction), in combination with antibiotic therapies that prevent bacterial overgrowth [71]. Other treatment strategies have been evaluated or are under study. Several randomized controlled trials demonstrated that, in contrast to standard-care control groups, SSc patients undergoing autologous hematopoietic stem cell transplantation (HSCT) have long-term survival advantages and clinical improvements [95]. The rationale behind stem cell therapy is that, after the depletion of immune cells, the reconstitution of immune cells with stem cell grafting can reestablish immune tolerance [96]. Despite these positive effects, however, there is an increased risk of transplant-related mortality. Moreover, although the autologous HSCT allows for clinical improvements related to the absence of pharmacological treatments, relapses may occur that necessitate the introduction of anti-rheumatic drugs. This is due to the development of the SSc disease, which involves and damages multiple internal organs. For this reason, the HSCT could be excluded for safety reasons and applied in a limited time-window [97]. Mesenchymal stromal cells (MSCs) may be an alternative to hematopoietic stem cells, with the potential to provide long-term benefits to patients with scleroderma [98]. These cells possess anti-inflammatory, antiproliferative, antifibrotic, and immunomodulatory properties. Furthermore, secreting different soluble molecules, they can directly repair damaged tissues. MSCs have a wide availability and a very low acute toxicity and, being very sensitive to their environment, they are able to modulate their activity according to the physio-pathological context in which they are located. MSCs for scleroderma treatment have been evaluated in different animal models that replicate SSc. In these studies, it is demonstrated that the injection of MSCs reduces collagen deposition, down-regulates TGF-beta expression and release, reduces skin and lung fibrosis, and decreases inflammation, Scl-70 autoantibodies, and oxidative markers. At the same time, MSCs increase matrix remodeling markers, which are responsible for the normal collagen arrangement in different tissues [97]. The use of MSCs has been tested in several clinical trials for autoimmune diseases, such as multiple sclerosis, Crohn’s disease and erythematous systemic lupus. There are few studies on SSc and many aspects of the disease still have to be addressed to ascertain their potential in association with other therapies.

Despite the large number of pharmacological treatments, SSc remains a significant medical challenge, with a high mortality and morbidity. While current therapies improve patients’ quality of life and survival, no curative treatment exists. Medical research studies and numerous clinical trials have attempted to identify new parameters involved in the SSc pathogenesis and to establish new promising diagnostic and therapeutic options.

MSCs display anti-fibrotic, angiogenic and immunomodulatory abilities, which might be of interest in the treatment of SSc, by acting on different processes that are dysregulated in this disease. Furthermore, when administrated locally or systemically, MSCs have clear beneficial effects in the reparative processes of injured tissues [99,100]. Since experimental studies showed that MSCs are incorporated only in small proportion in the injured tissues, it is possible that their beneficial effects are not due to the engraftment itself, but rather to the secretion of soluble mediators that act in a paracrine way. Soluble mediators produced by MSC exosomes are arousing great interest in relation to the development of novel cell-free therapeutic approaches.

## 4. Potential Role of Exosomes in Systemic Sclerosis

### 4.1. Biogenesis and Function of Exosomes

Exosomes are nanovesicles that range between 40–150 nm in diameter. They are surrounded by a phospholipid bilayer of endocytic origin, which is derived from multivesicular endosomes (MVEs), commonly called multivesicular bodies (MVBs). Exosome formation occurs when the membrane of the MVEs bulges inward and pinches off to create small membranous vesicles within the MVEs, which are packed with cytoplasmic contents, including proteins, RNAs, microRNA, and DNA. When MVEs fuse with the plasma membrane, exosomes are released into the extracellular environment [101]. Due to the mechanism of biogenesis, the exosomal membrane has the same orientation as the parental cell plasma membrane, and it is enriched in endosome-related membrane transport and fusion proteins, lipids, and tetraspanins. Since exosomes are released by almost all cells, both under physiological and pathological conditions, their molecular content can vary significantly [102]. The exact mechanisms involved in exosome packaging have not been fully elucidated, but they appear to be similar to the mechanisms involved in the packaging of lysosomal-bound MVEs, since endosomal-sorting complexes, required for transport (ESCRT) proteins, are found in exosomes [103]. Exosomes have mostly been characterized in immune cells (dendritic cells, B cells, T cell, and macrophages) and tumors [104,105]. First considered as a “waste bag” of cells [106], these vesicles have been considered to play important roles in intercellular communication and immune response [107,108,109,110]. These vesicles may exert their biological functions on target cells through receptor–ligand interactions, internalization by endocytosis, or vesicle-cell membrane fusion [111]. 

### 4.2. Exosomes and the Immune System

The immune system represents one of the earliest reported physiological targets of exosomes. The immunological activities of exosomes affect both innate and adaptive immunity and include antigen presentation, T cell activation and polarization in regulatory T cells, immunosuppression, and anti-inflammation activity. Lymphocytes B have been reported as the first immune cells able to secrete exosomes, which express abundant major histocompatibility complex (MHC) Class I and II molecules, B7.1 (CD80) and B7.2 (CD86), co-stimulatory molecules, and the intercellular adhesion molecule, ICAM-1 (CD54). Due to the presence of these proteins, the exosomes are able to activate CD4^+^ T cells and mediate antigen presentation by lymphocytes [112]. Different exosome populations have been found to be secreted by CD4^+^ T cells, and their release is differentially regulated by T-cell activation [113]. While some studies observed that exosomes can directly stimulate T cells in vitro, others have shown that these vesicles require antigen-presenting cells (APC) for this stimulation. The complex, peptide/MHC, carried by exosomes after their internalization [114] may be transported to the APC surface [115], or the antigenic peptide alone can be transferred to the cell membrane, which is near the MHC molecule that activates T lymphocytes [116]. Moreover, the number of MHCs and co-stimulatory molecules present on the exosome surface is critical for the formation of an effective immunologic synapse. 

Recent studies support the emerging role of exosomes in the immune tolerance–autoimmunity balance, highlighting their role in maintaining pathological autoimmune responses. For example, it has been reported that exosomes have a role in the progression of rheumatoid arthritis (RA) [117]. Indeed, synovial fibroblasts derived from patients with RA-release exosomes contain a membrane form of tumor necrosis factor, (TNF)-α, that is internalized by T cells, which activate AKT, and the nuclear factor kappa-light-chain-enhancer of activated B cells, becoming resistant to apoptosis. Moreover, exosomes isolated from the synovial fluids of RA patients contain citrullinated proteins, which are known to be autoantigens in this disease [118]. Interestingly, in the articular cartilage of osteoarthritis patients, a high level of exosomes containing annexin, which promotes mineral formation and the destruction of the articular chondrocytes, has been found [119]. It has also been found that patients with systemic lupus erythematosus have a higher level of exosomes in serum, compared to healthy controls. These vesicles were able to induce the production of pro-inflammatory cytokines (interferon--α, TNF-α, IL-1β, and IL-6) when co-cultured in vitro with peripheral blood mononuclear cells (PBMCs), isolated from healthy controls [120]. In the context of SSc, exosomes released by the different components of the immune system may affect the inflammation, which is the fundamental process behind the vasculopathy and in fibrosis. High levels of exosomes have been found in SSc patients, and their possible role has been postulated in the modulation of endothelial cell apoptosis, which is considered to be one primary pathogenic event in this disease [121,122]. On the other hand, exosomes can transport anti-inflammatory molecules to generate immunosuppressive effects. CD4^+^ T cell-derived exosomes have been described to inhibit CD8^+^ cytotoxic T-lymphocyte responses [123] and microRNA (miR) let-7d, contained in exosomes released by Treg cells, suppress Th1 cell proliferation and cytokine production [124]. 

### 4.3. Exosomes and Vasculopathy

A growing number of works have explored the role of exosomes in angiogenesis and vascular remodeling, especially in tumors [125]. PAH is often observed in SSc patients and is characterized by pulmonary vascular remodeling. The features of PAH are an increased proliferation, apoptosis resistance of pulmonary ECs, and vascular smooth muscle cells (VSMCs). The alternations between ECs and VSMCs leads to vascular thickening and stiffening, which results in a hemodynamic imbalance, with a low flow and high resistance [126]. Several studies recently reported an exchange of exosomes between pulmonary VSMCs and endothelial cells in PAH [127,128]. In a murine model of chronic hypoxia pulmonary hypertension, it was observed that exosomes secreted by pulmonary VSMCs were enriched in miR-143 and were up-taken by pulmonary ECs, inducing migration and angiogenesis. The genetic ablation and pharmacological inhibition of miR-143 prevented the development of PAH [127]. The intravenous delivery of MSC-derived exosomes in a mouse model of hypoxic pulmonary hypertension showed a cytoprotective action by inhibiting vascular remodeling and reducing pulmonary hypertension through the suppression of the signal transducer and activator of transcription 3 (STAT3) and up-regulation of the miR-17 superfamily. In vitro experiments also confirmed the anti-inflammatory role of exosomes produced by MSC in pulmonary ECs [128]. 

### 4.4. Exosomes and Fibrosis

Fibrosis is a reparative or reactive process that is characterized by the formation and deposition of excess fibrous connective tissue, with progressive architectural remodeling in nearly all tissues and organs. While in the short term, this fibrogenic response may have adaptive features, when it is sustained over a prolonged period of time, it causes parenchymal scarring, cellular dysfunction, and organ failure [129]. Four major phases of the fibrogenic response have been proposed: 1) The initiation of the response: driven by a primary injury to the organ; 2) the activation of effector cells; 3) the elaboration of the extracellular matrix; and 4) the dynamic deposition of the extracellular matrix, with the progression of fibrosis and organ failure. Several studies have provided evidence relating to different fibrotic diseases that suggests that stressed parenchymal cells can release exosomes that transmit stress signals to neighboring immune cells, such as macrophages, inducing a fibrotic process of several organs [130,131,132,133]. 

In the past, it has been shown that transplanting the human umbilical cord-MSCs (hucMSCs) into the fibrotic liver ameliorates fibrosis and restores liver function. Tingfen Li et al. demonstrated in vivo that the resolution of liver fibrosis is linked to the paracrine effect of exosomes isolated from hucMSCs (hucMSC-Ex), which reduce hepatic inflammation and collagen deposition [134]. Zhang J. and co-workers demonstrated that exosomes derived from MSCs can play important roles in repairing injured tissue [135]. In particular, they showed that exosomes isolated from MSCs derived from pluripotent stem cells (hiPSC-MSC-Exos) promote cutaneous wound healing, collagen synthesis, and vascularization at wound sites in a rat full-thickness skin defect model. Moreover, hiPSC-MSC-Exos promote the synthesis of collagen by fibroblasts and the formation of new vessels by human umbilical vein endothelial cells (HUVEC) in vitro [135]. In support of these observations, it has been found that exosomes derived from adipose MSCs (ASCs-Exos) can be internalized by fibroblasts to stimulate cell migration, proliferation, and collagen synthesis. Furthermore, the internalization of these vesicles increases the gene expression of N-cadherin, cyclin-1, proliferating cell nuclear antigens, and collagen I and III in fibroblasts [136]. In vivo tracing experiments showed that ASCc-Exos can be recruited and sent to the wound area in a mouse skin incision model and accelerate cutaneous wound healing. The systemic administration of exosomes in the early stages of wound healing caused an increase in collagen I and III production, while an inhibitory effect on collagen expression in reducing scar formation has been observed when exosomes were administrated in the late stage [136]. As reported above, the TGF-β/SMAD pathway is considered the main signal that is deregulated in SSc fibroblasts and contributes to the initiation and development of the pathological fibrotic response [8]. High levels of TGF-β activity at the wound/inflammatory site have been associated with excessive scarring and fibrosis. For this reason, interfering with the activity of the TGF-β/SMAD2 signaling might suppress myofibroblast differentiation and aggregation, reducing excessive fibrosis and scar formation [137]. In particular, it has been observed that umbilical cord-derived (u)-MSC exosomes were enriched in microRNAs (miR-21, miR-23a, miR-125b and miR-145), which play a key role in suppressing myofibroblast formation through the inhibition of excess alpha-SMA and collagen deposition, associated with the activity of the TGF-β/Smad2 pathway.

## 5. Exosomes: New Perspective for the Diagnosis and Treatment of Systemic Sclerosis 

The properties of exosomes have stimulated extensive research relating to the exploitation of them as a source of biomarkers for the diagnosis and the follow-up of several pathologies. Indeed, exosome transport proteins, miRNAs, and nucleic acids may vary under pathological conditions or during pharmacological treatment. Moreover, their presence in several biological fluids makes them ideal candidates for simple and non-invasive analysis. Studies that measure and characterize exosomes in subjects affected by SSc are very limited and very little is known about their specific role in this rheumatic disease. In 2016, Nakamura K. and collaborators investigated the putative role of exosomes in wound healing in SSc. They found that the expression of exosome markers, CD63, CD9 and CD81, in skin samples of patients with dcSSc and lcSSc was increased, compared with control samples (NS) [138]. Furthermore, they showed in vitro that the mRNA levels of exosome markers, CD63, CD9, and CD81, and exosome secretion were increased in SSc dermal fibroblasts, compared to normal fibroblasts. The in vitro addition of conditioned media from the SSc-fibroblast (containing exosomes) to NS fibroblasts induced collagen type I alpha 1 chain (COL1A1) and collagen type I alpha 2 chain (COL1A2) mRNA levels. The analysis of the exosome level in serum samples of SSc patients revealed a significant decrease in both dcSSc and lcSSc patients, compared to that in NS. Intriguingly, patients with decreased serum exosome levels had vascular involvements, such as skin ulcers or pitting scars. This exosome reduction in the serum of SSc patients with vascular involvements is probably related to the disturbed transfer of exosomes from the skin tissue to the blood stream, which in turn causes a delay of wound healing by the down-regulation of type I collagen, resulting in a higher susceptibility to skin ulcers and pitting scars. The authors also investigated the effect of serum-derived exosome supplementation on cutaneous wound healing in mice, demonstrating that the diameter of wounds was smaller in mice treated with exosomes, suggesting a role of these vesicles in wound healing induction [138]. Another work by Guiducci and co-authors investigated the possible relationship between plasma exosomes and disease manifestation in patients with SSc [139]. The authors isolated and analyzed blood exosomes from 37 patients with SSc, compared to 15 healthy subjects, and found that the total number of these particles was increased in patients with SSc. Moreover, significant increases were found for exosomes derived from platelets, ECs, monocytes, and T cells, reflecting the involvement of these types of cells in SSc. Furthermore, patients with cutaneous ulcers showed a significantly reduced total number of vesicles, compared to patients without cutaneous ulcers, suggesting a possible role in the pathogenesis of this disease [139]. Recently, Wermuth P.J. and co-workers proposed a crucial role of exosomes in the paracrine modulation of molecular programs in neighboring and distant normal cells, as follows: Released exosomes enter the circulation and fuse with distant target cells, inducing a profibrotic molecular program that sustains the SSc-etiologic factor and spreads the pathogenic mechanisms for SSc propagation. The authors analyzed exosomes isolated from the serum of SSc patients and tested the ability of exosomes to induce a profibrotic phenotype in normal human dermal fibroblast in vitro [140]. Several miRNAs implicated in the stimulation of profibrotic target genes or inhibition of antifibrotic genes were significantly increased in exosomes isolated from SSc serum, compared to normal subjects. In particular, four profibrotic miRNAs (let-7g, miR-17, miR-23b and miR-29a) were increased 5-fold, and four antifibrotic miRNAs (let-7a, miR125b, miR140 and miR146a) were decreased 70-fold in exosomes isolated from SSc serum (Table 1). The analysis of the literature reveals that the expression levels of some of the deregulated exosomal miRNAs, identified by Wermuth and coworkers, are altered in SSc patients. For example, let-7 family microRNA in skin was lower in SSc patients with pulmonary hypertension (PH) than in those without PH [141]. let-7a resulted in the down-regulation of the skin of SSc patients, compared to controls, and it is implicated in collagen I regulation [142]. miR-29a was down-regulated in the skin and dermal fibroblasts of SSc patients, compared to controls, and it has been implicated in the regulation of collagen expression and apoptosis [143,144,145]. Polymorphisms, rs2910164C/G, of miR-146a were associated with lung fibrosis development, telangiectasia, and SSc onset [146,147]. Wermuth et al. also evaluated the effects of isolated serum exosomes on the gene expression and biosynthetic activity of normal human dermal fibroblasts [140]. Exosomes isolated from the serum of SSc patients stimulated the expression of genes encoding extracellular matrix components, such as COL1A1, collagen type III alpha 1 chain and fibronectin-1. Moreover, the strongest up-regulation was mediated by serum exosomes from diffuse SSc patients, while a moderate up-regulation was mediated by exosomes isolated from limited SSc donor sera. Serum exosomes from both subsets of SSc patients also induced dose-dependent increases in the expression of genes associated with myofibroblast activation (cartilage oligomeric matrix protein, actin alpha 2, smooth muscle and the fibronectin extra domain A), TGF-β, and connective tissue growth factor genes. Recently, miR-151-5p has been suggested as a promising biomarker, both for diagnosis and treatment of SSc. Chen C. and co-workers showed that the miR-151-5p contained in exosomes, released from donor MSCs, was transferred to recipient bone marrow MSCs in an SSc mice model. The up-take of exosomes containing these miRNAs by target cells led to the inhibition of IL4Rα expression and down-regulation of the mTOR pathway, increasing osteogenic differentiation and reducing adipogenic differentiation. Furthermore, they showed that the systemic delivery of miR-151-5p caused the rescue of osteopenia, impaired bone marrow MSCs, tight skin, and immune disorders in SSc mice [148]. Another aspect that makes exosomes particularly interesting is their use as a vehicle for the administration of biological or therapeutic compounds. Indeed, they have intrinsic cell targeting properties, are stable in circulation and are recognized as a “self” by the immune system. Among the most fascinating applications of the exosomes in nano-delivery is their use as carriers of specific therapeutic miRNAs. In 2016, Wang and collaborators engineered MSCs to overexpress miR-let7c in exosomes and showed that these nano-particles were selectively internalized by damaged kidney cells, resulting in the attenuation of the kidney injury and reduction of fibrosis [149]. Adipose-derived MSCs (ADSCs) were engineered to overexpress miRNA-181-5p (miR-181-5p-ADSCs) in order to selectively home exosomes to mouse hepatic stellate (HST-T6) cells or a CCl4-induced liver fibrosis murine model. In vitro analysis confirmed that the transfer of miR-181-5p from miR-181-5p-ADSCs occurred via a secreted exosomal uptake, leading to the down-regulation of fibrotic genes induced by TGF-β1 and reducing fibrosis, outlining a possible therapeutic use of engineered exosomes in liver fibrosis [150]. MiR-1343 has been described as a potent repressor of TGF-β-signaling and fibrosis through the direct repression of TGF-β receptors in primary fibroblasts and lung epithelial cell lines [151]. It has been shown that exosomes containing miR-1343 can be picked up by primary lung fibroblasts and consequently reduce TGF-β signal transduction and markers of fibrosis [152]. 

## 6. Conclusions

SSc is a systemic autoimmune disease of unknown etiology, which is often fatal and is characterized by a fibrotic process that effects the skin, microvasculature, and internal organs. The pathogenesis is complex, little understood, and involves the interplay of several cellular components, mediators, and molecular pathways. A lack of curative therapies often leads to severe disability and death in SSc patients. Moreover, the identification of specific and validated, non-invasive bio markers for diagnosis or to follow the course of the disease is still lacking. Exosomes are membrane-surrounded nano-vesicles of endocytic origin, which are produced by several types of cells and implicated in inter-cellular communication, containing nucleic acids, proteins, and lipids. As a consequence of their ability to transport messages over short and long distances, and since they contain molecules capable of influencing numerous biological processes in the target cell, exosomes represent a fascinating field of study in relation to SSc. Even if their specific role in the etiopathology of SSc remains unknown, their ability to connect different cell types and different parts of the body could represent an intriguing starting point for the molecular characterization of this complex multi-event and multi-organ pathology (Figure 1). Immune cells produce exosomes that can act on the same immune cells or on the cells of other tissues, such as endothelial cells regulating, for example, the inflammatory processes. MSC-derived exosomes have immunomodulatory properties, contain miRNAs with anti- or profibrotic activity, and regulate the formation of new vessels. Exosomes secreted by vascular smooth muscle cells are up-taken by ECs, influencing the migratory properties of target cells and angiogenesis. Exosomes produced by several types of cells, such as ECs, fibroblasts, and immune cells, can contain miRNAs and other molecules that can be picked up by several target cells to regulate different processes, which lead to the manifestation of the main symptoms of SSc.

Another interesting and related point is the identification and characterization of microRNAs and of proteins transported by the exosomes and deregulated in pathological conditions during disease progression or during drug treatment, which could represent a reserve of putative biomarkers for the clinical management of SSc patients. Furthermore, exosomes can be isolated from different biological fluids, and this makes them particularly interesting for the establishment of non-invasive procedures. Thanks to their intrinsic cell targeting properties, stability in circulation, and bio-compatibility, they are emerging as new promising vehicles, both for drugs and biological therapeutics. Gene therapy studies are underway to evaluate the efficacy of these nano-vesicles in transporting certain molecules or drugs to specific sites. Moreover, their important role in intercellular communication makes exosomes particularly interesting in defining the mechanisms underlying the onset and progression of a systemic and complex pathology, such as SSc, which involves numerous districts, several organs, different cell types, and a complex interaction with the microenvironment and the extracellular matrix. Further research is still needed to clarify the mechanisms underlying SSc and to find an appropriate therapy, but the study of the content and role of these messengers opens fascinating and hopeful scenarios for precision medicine for the treatment of SSc. 

## Figures and Tables

**Figure 1 ijms-20-04337-f001:**
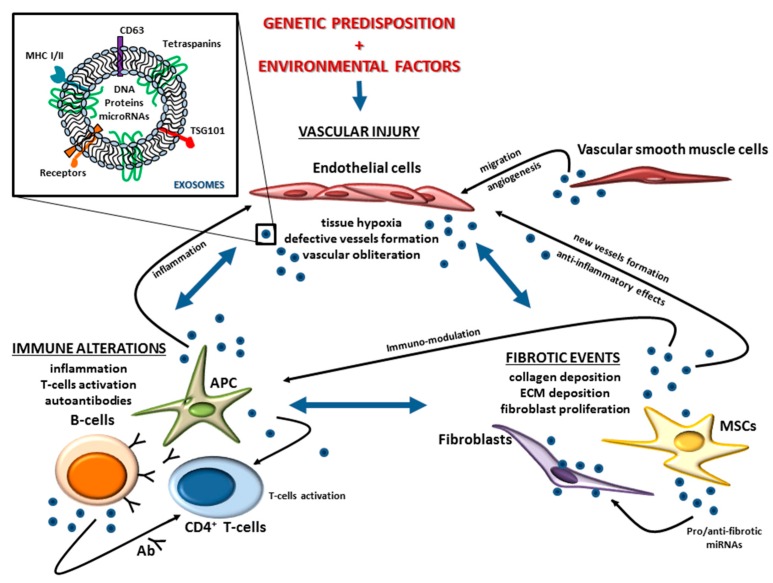
Possible role of exosomes in SSc: Messengers between immune, vascular, and fibrotic components. Genetic predisposition and environmental factors contribute to the onset of SSc. SSc is characterized by a progressive and systemic multi-organ fibrosis, which is preceded by vessel vasculopathy, inflammation, and autoimmunity. Exosomes, membrane-surrounded nano-vesicles, are produced by several types of cells (immune cells, endothelial cells, MSCs, fibroblasts, and vascular smooth muscle cells) and are implicated in inter-cellular communication. Exosomes can carry several types of molecules, i.e., DNA, microRNA, and proteins, which are able to regulate different processes in the target cell, thus influencing its behavior. Immune cells produce exosomes that can act on the same immune cells or on the cells of other tissues, such as endothelial cells regulating, for example, the inflammatory processes. MSCs-derived exosomes have immune-modulatory properties, contain miRNAs with anti- or profibrotic activity and regulate the formation of new vessels. Exosomes secreted by vascular smooth muscle cells are up-taken by endothelial cells, influencing the migratory properties of target cells and angiogenesis. Blue arrows mean interconnection between vasculopathy, autoimmunity, and fibrosis; black arrows mean exosomes affecting target cells.

**Table 1 ijms-20-04337-t001:** Comparison of the exosomal miRNAs identified by Wermuth PJ et al. in the serum of SSc patients and their possible role in systemic sclerosis.

Serum Exosomal miRNAs in SSc Patients [137]	Level of Expression in Serum Exosomes [137]	Site of Expression in SSc Patients	Level of Expression in SSc Patients	Role in SSc Patients	Reference
**Let-7g**	Up-regulated	Skin	Down-regulated	Negative correlation with pulmonary hypertension	[141]
**miR-17**	Up-regulated	----	----	----	----
**miR-23b**	Up-regulated	----	----	----	----
**miR-29a**	Up-regulated	Skin, fibroblasts	Down-regulated	Regulation of collagen expression	[143]
		Dermal fibroblasts	Down-regulated	Repression of TAB1 and prevention of fibrosis	[144]
		Dermal fibroblasts	Down-regulated	Induction of apoptosis	[145]
**Let-7a**	Down-regulated	Skin fibroblasts, serum	Down-regulated	Inhibition of collagen I expression	[144]
**miR-125b**	Down-regulated	----	----	----	----
**miR-140**	Down-regulated	----	----	----	----
**miR-146a**	Down-regulated	PBMC	Down-regulated	Lung fibrosis development	[146]
		Skin (Rs2910164 C/G polymorphism)	----	Associated with telangiectasia	[147]

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
