# Peer review of "Exosomes in Systemic Sclerosis: Messengers Between Immune, Vascular and Fibrotic Components?"

_ijms, 2019, doi:10.3390/ijms20184337_

Round 1
Reviewer 1 Report
The review manuscript by Colletti et. al. describes the pathogenesis of systemic sclerosis (SSc) and the major role played by exosomes in mediating SSc. The authors also speculate that targeting exosomes could be of therapeutic benefits in SSc patients. Though the manuscript highlights some of the major aspects of SSc, there are few concerns:
1. The language of the manuscript is poor. There are a lot of grammatical errors and many phrases throughout the manuscript that do not make any sense or are wrongly structured. This makes the review manuscript very hard to follow and understand.
2. Some of the abbreviations are not well defined. Such as Th, M2, etc.
3. The authors have given an elaborate description of fibrosis rather than giving it to SSc. They introduced the important role of the innate and adaptive arm of the immune system in mediating SSc, but the details are missing in the main text.
4. The authors have described the various functions of exosomes in SSc, but they never clearly explained how the exosomes connect the immune, vascular and fibrotic components, which is also the title of the manuscript. I feel that because the authors elaborately described each component in detail they failed to link these components together which makes the reader hard to follow up the events occurring in SSc.
Author Response
Dear Reviewer 1,
Thank you for your reply regarding our manuscript ijms-562265 entitled “Exosomes in Systemic Sclerosis: messengers between immune, vascular and fibrotic components”. We are grateful for reviewer’s comments and we appreciate the opportunity that we have been given to further revise our manuscript.
Please find below our revisions point by point according to the reviewers ‘comments. All the changes have been reported in the manuscript in red.
The language of the manuscript is poor. There are a lot of grammatical errors and many phrases throughout the manuscript that do not make any sense or are wrongly structured. This makes the review manuscript very hard to follow and understand.
The revised manuscript has been submitted for Rapid grammar check (regular edit) in the MDPI author services and English editing (English Editing ID: english-11537)
Some of the abbreviations are not well defined. Such as Th, M2, etc.
Abbreviations have been checked in the main text and spelled out the first time a word is mentioned when it is mentioned more than once. The list of abbreviations has been updated with all the missing ones.
The authors have given an elaborate description of fibrosis rather than giving it to SSc. They introduced the important role of the innate and adaptive arm of the immune system in mediating SSc, but the details are missing in the main text.
Some details regarding the innate and the adoptive arm of the immune system in mediating SSc have been added in the paragraph “2.1. Immune system abnormalities”. In particular, the following sentences have been inserted in the main text:
B cells are considered to be involved mainly in autoantibody production, even if there is no evidence proving the direct causal significance of SSc autoantibodies in the disease. (lanes 71-72 manuscript version: “ijms-562265-revised and edited”); Moreover, B cells seem to be also involved in the production of interleukin (IL)-6 and profibrotic mediators, and importantly, they act as antigen-presenting cells for T cells. (lanes 83-85 manuscript version: “ijms-562265-revised and edited”); Type 2 immune responses involve other innate players, such as innate lymphoid type 2 cells, M2c CD163+ macrophages, basophils, mast cells, eosinophils, and epithelial cells producing IL-33 [23]. Based on an interesting hypothesis, the clonal expansion of sel-reactive CD4+ cytotoxic T cells could be responsible for “cytolic fibrosis,” characterized by an induced apoptosis of normal tissue, production of profibrotic cytokines, in particular TGF-β and IL-1β, finally leading to abnormal tissue repair and fibrosis [23]. (lanes 99-105 manuscript version: “ijms-562265-revised and edited”);
The authors have described the various functions of exosomes in SSc, but they never clearly explained how the exosomes connect the immune, vascular and fibrotic components, which is also the title of the manuscript. I feel that because the authors elaborately described each component in detail they failed to link these components together which makes the reader hard to follow up the events occurring in SSc.
A further explanation of how exosomes could be mediators and messagers that travel between the three main components (immune, vascular and fibrotic components) at the base of SSc has been added to the paragraph “Conclusion”. In particular we have written:
Immune cells produce exosomes that can act on the same immune cells or on the cells of other tissues, such as endothelial cells regulating, for example, the inflammatory processes. MSC-derived exosomes have immunomodulatory properties, contain miRNAs with anti- or profibrotic activity and regulate the formation of new vessels. Exosomes secreted by vascular smooth muscle cells are up-taken by ECs, influencing the migratory properties of target cells and angiogenesis. Exosomes produced by several types of cells, such as ECs, fibroblasts, and immune cells, can contain miRNAs and other molecules that can be picked up by several target cells to regulate different processes, which lead to the manifestation of the main symptoms of SSc. (lanes 539-547 manuscript version: “ijms-562265-revised and edited”)
There is no clear evidence of how exosomes can correlate these three aspects within a complex pathology such as SSc, but clear evidence highlights their involvement in each aspect which suggests their possible role as components that interconnect different cell types and various organs. To emphasize the possibility of their involvement we have added a question mark to the title that has now become “Exosomes in Systemic Sclerosis: messengers between immune, vascular and fibrotic components?” (lane 3 manuscript version: “ijms-562265-revised and edited”)
Moreover:
-we added following references:
[23] Pillai, S. T and B lymphocytes in fibrosis and systemic sclerosis. Curr Opin Rheumatol. 2019, Jul 31. [82] Baqir, M.; Makol, A.; Osborn, T.G.; Bartholmai, B.J.; Ryu, J.H. Mycophenolate mofetil for scleroderma-related interstitial lung disease: A real world experience. PLoS One. 2017, 12, e0177107. [83] Koutroumpas, ; Ziogas, A.; Alexiou, I.; Barouta G.; Sakkas L.I. Mycophenolate mofetil in systemic sclerosis-associated interstitial lung disease. Clin Rheumatol. 2010, 29, 1167-1168. [84] Shenoy, P.D.; Bavaliya, M.; Sashidharan, S.; Nalianda, K.; Sreenath, S. Cyclophosphamide versus mycophenolate mofetil in scleroderma interstitial lung disease (SSc-ILD) as induction therapy: a single-centre, retrospective analysis. Arthritis Res Ther. 2016, 18, 123. [85] Khanna, D.; Denton, C.P.; Jahreis, A.; van Laar, J.M,; Frech, T.M., Anderson, M.E.; et al. Safety and efficacy of subcutaneous tocilizumab in adults with systemic sclerosis (faSScinate): a phase 2, randomised, controlled trial. Lancet. 2016, 387, 2630-2640.
-all references in the main text and in table 1 have been re-numbered appropriately.
Best Regards,
Dr. Cristina Antinozzi
Reviewer 2 Report
Dear Authors,
This article is the review of the relationship between exosome and systemic sclerosis (SSc) including immune, vascular and fibrotic systems. Exosome includes abundant and various protein, DNA, mRNA and microRNA. This review article mentioned well about importance of exosome analysis in SSc.
I have several questions and need author's comments, below;
In the review, immune system abnormalities in SSc were mentioned. Do you need to mention about treatment using Mycophenolate mofetil (MMF) and tocilizumab? Both medications have been tried to treat SSc with or without lung involvement. Several major genome wide meta analysis were performed in the world. In those meta analysis, is there anything relationship between disease related loci and microRNA?
Sincerelly yours,
Author Response
Dear Reviewer 2,
Thank you for your reply regarding our manuscript ijms-562265 entitled “Exosomes in Systemic Sclerosis: messengers between immune, vascular and fibrotic components”. We are grateful for reviewer’s comments and we appreciate the opportunity that we have been given to further revise our manuscript.
Please find below our revisions point by point according to the reviewers ‘comments. All the changes have been reported in the manuscript in red.
In the review, immune system abnormalities in SSc were mentioned. Do you need to mention about treatment using Mycophenolate mofetil (MMF) and tocilizumab? Both medications have been tried to treat SSc with or without lung involvement.
We mentioned the treatment with MMF and tocilizumab in the paragraph “Treatments of Systemic Sclerosis” and we added the following sentences:
Mycophenolate mofetil (MMF), an inactive prodrug of mycophenolic acid, which inhibits inosine monophosphate dehydrogenase and the proliferation of T and B cells, has been tested for its capacity to treat SSc-associated ILDs. Several studies have shown that MMF improved the skin score and stabilized pulmonary function of patients [82–84] Tocilizumab (TCZ), a humanized monoclonal IgG1 antibody that acts by blocking the signals mediated by IL-6, has been proposed as a possible treatment for some manifestations of SSc, including ILD. Interestingly, a recent open-label pilot study showed a skin score improvement and lung function stabilization after TCZ treatment [85]. (lanes 248-255 manuscript version: “ijms-562265-revised and edited”);
Several major genome wide meta analysis were performed in the world. In those meta analysis, is there anything relationship between disease related loci and microRNA?
Literature analysis shows that between miRNAs cited in the main text, systemic sclerosis disease susceptibility could be associated with the polymorphism rs2910164C/G of miR-146a [Vreca, M. et al., Immunol Lett. 2018, 204, 1-8; Sakoguchi, A. et al., Clin Exp Dermatol. 2013, 38, 99-100]. We specified this in the main text and we added the polymorphism (lane 488 manuscript version: “ijms-562265-revised and edited”) and the word “onset” (lane 489 manuscript version: “ijms-562265-revised and edited”).
Moreover:
-we added following references:
[23] Pillai, S. T and B lymphocytes in fibrosis and systemic sclerosis. Curr Opin Rheumatol. 2019, Jul 31. [82] Baqir, M.; Makol, A.; Osborn, T.G.; Bartholmai, B.J.; Ryu, J.H. Mycophenolate mofetil for scleroderma-related interstitial lung disease: A real world experience. PLoS One. 2017, 12, e0177107. [83] Koutroumpas, ; Ziogas, A.; Alexiou, I.; Barouta G.; Sakkas L.I. Mycophenolate mofetil in systemic sclerosis-associated interstitial lung disease. Clin Rheumatol. 2010, 29, 1167-1168. [84] Shenoy, P.D.; Bavaliya, M.; Sashidharan, S.; Nalianda, K.; Sreenath, S. Cyclophosphamide versus mycophenolate mofetil in scleroderma interstitial lung disease (SSc-ILD) as induction therapy: a single-centre, retrospective analysis. Arthritis Res Ther. 2016, 18, 123. [85] Khanna, D.; Denton, C.P.; Jahreis, A.; van Laar, J.M,; Frech, T.M., Anderson, M.E.; et al. Safety and efficacy of subcutaneous tocilizumab in adults with systemic sclerosis (faSScinate): a phase 2, randomised, controlled trial. Lancet. 2016, 387, 2630-2640.
-all references in the main text and in table 1 have been re-numbered appropriately.
-The revised manuscript has been submitted for Rapid grammar check (regular edit) in the MDPI author services and English editing (English Editing ID: english-11537)
Best Regards,
Dr. Cristina Antinozzi
Round 2
Reviewer 1 Report
I thank the authors for answering all my queries.